# Secure Cyber Defense: An Analysis of Network Intrusion-Based Dataset CCD-IDSv1 with Machine Learning and Deep Learning Models

**Niraj Thapa [1], Zhipeng Liu [2], Addison Shaver [2], Albert Esterline [2], Balakrishna Gokaraju [1] and Kaushik Roy [2,*]**

[1] Department of Computational Data Science and Engineering, North Carolina A&T State University, Greensboro, NC 27411, USA; nthapa@aggies.ncat.edu (N.T.); bgokaraju@ncat.edu (B.G.)

[2] Department of Computer Science, North Carolina A&T State University, Greensboro, NC 27411, USA; zliu2@aggies.ncat.edu (Z.L.); awshaver@aggies.ncat.edu (A.S.); esterlin@ncat.edu (A.E.)

\* Correspondence: kroy@ncat.edu

**Abstract:** Anomaly detection and multi-attack classification are major concerns for cyber defense. Several publicly available datasets have been used extensively for the evaluation of Intrusion Detection Systems (IDSs). However, most of the publicly available datasets may not contain attack scenarios based on evolving threats. The development of a robust network intrusion dataset is vital for network threat analysis and mitigation. Proactive IDSs are required to tackle ever-growing threats in cyberspace. Machine learning (ML) and deep learning (DL) models have been deployed recently to detect the various types of cyber-attacks. However, current IDSs struggle to attain both a high detection rate and a low false alarm rate. To address these issues, we first develop a Center for Cyber Defense (CCD)-IDSv1 labeled flow-based dataset in an OpenStack environment. Five different attacks with normal usage imitating real-life usage are implemented. The number of network features is increased to overcome the shortcomings of the previous network flow-based datasets such as CIDDS and CIC-IDS2017. Secondly, this paper presents a comparative analysis on the effectiveness of different ML and DL models on our CCD-IDSv1 dataset. In this study, we consider both cyber anomaly detection and multi-attack classification. To improve the performance, we developed two DL-based ensemble models: Ensemble-CNN-10 and Ensemble-CNN-LSTM. Ensemble-CNN-10 combines 10 CNN models developed from 10-fold cross-validation, whereas Ensemble-CNN-LSTM combines base CNN and LSTM models. This paper also presents feature importance for both anomaly detection and multi-attack classification. Overall, the proposed ensemble models performed well in both the 10-fold cross-validation and independent testing on our dataset. Together, these results suggest the robustness and effectiveness of the proposed IDSs based on ML and DL models on the CCD-IDSv1 intrusion detection dataset.

**Keywords:** intrusion detection system; CCD-IDSv1; machine learning; deep learning; KNN; CART; RF; XGBoost; CNN; LSTM; ensemble

## 1. Introduction

Cyber defense involves anticipating adversarial actions to counter intrusions. It is crucial to secure this infrastructure, which has deeply penetrated nearly all aspects of our lives, including social, economic, and political systems. Over the years, researchers have developed robust Intrusion Detection Systems (IDSs) to secure cyberspace. Secured cyber defense can lead to normal operations by organizations that reach a certain threshold while facing persistent threats and sophisticated attacks. It is critical to develop a proactive secured cyber defense system with these ever-evolving technologies.

Proactive intrusion detection monitors a network for malicious activity and optimizes itself with any new information it learns. It differs from the traditional firewall, which is based upon a static set of rules. Signature-based and anomaly-based are some typical

IDSs [1]. The signature-based detection method differs from the anomaly-based detection method in terms of its static nature. Hence, anomaly-based IDSs that can act against unknown attacks are preferred over signature-based IDSs.

Machine learning (ML) and deep learning (DL) models are trained for normal and anomalous activities to build anomaly-based IDSs. Classical supervised ML models are white-box models, which allow for the interpretability of features, whereas DL models are black-box models focused more on higher performance metrics. Feature importance can be generated from ML models, which can assist in further feature engineering. Hence, both ML and DL models are implemented in this research work to account for both performance and interpretability.

Current intrusion-based datasets need to be updated with ever-growing new threats. Furthermore, the development of a robust and optimized ML or DL model on those datasets is required. So, both the development of datasets and optimized ML/DL models built on those datasets are the keys to develop effective IDSs.

In the research reported in this paper, we develop a labeled flow-based dataset, Center for Cyber Defense (CCD)-IDSv1, for the evaluation of anomaly-based IDSs. The dataset is generated by emulating a small network environment using OpenStack. Furthermore, Argus is used to generate different network flow-based features. Finally, we implement both ML and DL models on this dataset to create robust IDSs. The contribution of this research can be summarized as: (1) data collection, (2) comparative analysis using different ML and DL models, (3) development of two ensemble models, and (4) feature importance.

The rest of the paper is organized as follows. Section 2 includes a literature review. Section 3 covers materials and methods, which include development and description of the dataset, preprocessing, and different ML and DL models. Section 4 presents results and analysis, and Section 5 is discussion and conclusions.

## 2. Related Work

IDSs can track incoming and outgoing network traffic to detect network anomalies based on training on the anomaly-based dataset. Several publicly available datasets, including KDD cup 99 [1], have been used extensively for the evaluation of IDSs. However, most of the publicly available datasets may not contain attack scenarios based on evolving threats. The KDD99 dataset [1] is one of the widely used network intrusion datasets for binary classification [2]. The KDD99 dataset was created from DARPA network dataset files [3]. Different ML models have been used for the classification of this dataset, with the random forest (RF) classifier attaining the highest accuracy for detecting and classifying all KDD99 dataset attacks [4]. A deep neural network has been applied to this dataset, which gave high accuracy as well [5]. Chowdhury et al. [6] implemented a few-shot intrusion detection system using support vector machines (SVM) and K-Nearest Neighbor (KNN) on features extracted from a trained deep convolutional neural network (CNN) for intrusion detection.

However, the analysis of the KDD99 dataset shows a high percentage of duplicates in both training and test datasets. These duplications increase biasness, thus hampering the model from attaining generalizability. Tavallaee et al. [2] developed the NSL-KDD dataset by removing the redundant information from the KDD99 dataset and by optimizing it. Revathi et al. [7] and Ever et al. [8] developed IDSs based on the NSL-KDD dataset using different ML models with high test accuracies. Su et al. developed a DL-based model, BAT [9], combining bidirectional LSTM and attention mechanisms, which was trained on the NSL-KDD dataset. Both KDD99 and NSL-KDD are comparatively old and could be outdated for the present intrusion threats.

An IP flow-based intrusion detection dataset has been developed by Sperotto et al. [10] with intrusion information only, and Shiravi et al. [11] included normal user behavior as well. However, due to its lack of external servers, it offers low practicality. Ring et al. [12,13] developed a labeled flow-based dataset, the Coburg Intrusion Detection Data Set (CIDDS). It contains attacks such as denial of service, brute force, and port scans. Verma et al. [14]

developed an IDSs and applied KNN and K-means clustering on the CIDDS-001 dataset for performance evaluation.

Sharafaldin et al. developed the CICIDS2017 [15] dataset, which contains benign and up-to-date common attacks resembling true real-world data. It includes the results of the network traffic analysis using CICFlowMeter and labeled flow based on the time stamp, source and destination IPs, source and destination ports, protocols, and attacks. However, CICFlowMeter can extract around 80 network flow-based features and does not include source/destination bits per second and record byte offset in file or stream. In this research work we use Argus, which can extract 127 network flow-based features.

Shurman et al. [16] proposed two models in an attempt to detect anomalies on the CICDDoS2019 dataset [17]. The first model was a hybrid model that encompasses signature-based method with an anomaly-based method. The second model was an LSTM model. However, the work only attempted to detect a specific DoS attack, and the methods were not applied on various datasets.

In this research, we develop a labeled flow-based network intrusion dataset CCD-IDSv1 with five attacks and a higher number of network features. We then build IDSs based on our developed dataset using different ML and DL models. We perform both binary and multi-class classification for anomaly detection and multi-attack classification, respectively. We develop an ensemble model to improve IDSs performance. Furthermore, feature importance is also studied for the CCD-IDSv1 dataset.

## 3. Materials and Methods

In this section, we present the development of an intrusion detection dataset and describe the different ML and DL models used.

### 3.1. CCD-IDSv1 Dataset

The CCD-IDSv1 dataset was developed in the CCD lab at North Carolina A&T State University for the evaluation of an anomaly-based network intrusion detection system. It is a labeled flow-based dataset containing NetFlow data. The dataset is generated by emulating a small network environment using OpenStack.

#### 3.1.1. Emulated Network Environment

OpenStack is a cloud operating system that controls large pools of computing, storage, and networking resources throughout a data center, all managed and provisioned through APIs with common authentication mechanisms.

The network environment, as shown in Figure 1, is implemented on OpenStack. Two internal networks, internal_1 and internal_2, are created. Five instances of Operating System are created for each internal network, respectively. For this research work, Linux environments, Kali and Ubuntu, are used. Kali Linux is primarily used as a penetration testing environment that contains different attacks by default. On the other hand, Ubuntu is one of the most widely used Linux systems in the world. Five different Kali systems are used to attack five different Ubuntu systems in parallel.

#### 3.1.2. Generation of Malicious and Normal Traffic

This research contains five different attacks: man-in-the-middle (MITM), address resolution protocol (ARP) Denial of Service (DoS), user datagram protocol (UDP) Flood DoS, Hydra Bruteforce, and Slowloris. Table S1 (supplementary materials) contains further details of these attacks. Our setting allows one to add more attacks easily. These attacks are carried out by Kali systems on Ubuntu systems at random time intervals. Every Kali system will infiltrate the Ubuntu system.

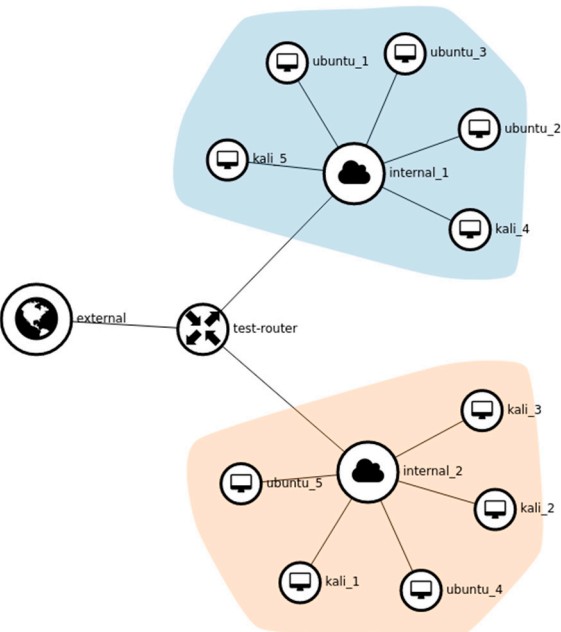

**Figure 1.** OpenStack network environment.

Wireshark is installed in every Ubuntu system that is used to collect the network flow information. Wireshark listens to the network traffic in and out of the Ubuntu systems and collects the network flow information in raw PCAP format. Wireshark is compatible with both Windows and Unix systems. This way, malicious traffic is generated.

Automated script imitating general usage by a person is used to create normal traffic. The Bash script is used to imitate general usage with randomized web browsing, video streaming, and file downloads. Wireshark is used to collect the network flow information for normal traffic as well in Ubuntu system. The dataset was collected for total of seven days. Table 1 shows the total number of instances for all attacks and normal usage.

**Table 1.** Total instances for different attacks and normal usage.

| Dataset | Instances |
|---------|-----------|
| Normal usage | 445,163 |
| MITM | 49,394 |
| ARP DoS | 33,440 |
| UPD Flood DoS | 45,003 |
| Hydra bruteforce | 2,802,864 |
| Slowloris | 4,232,788 |

### 3.1.3. Feature Extraction and Labeling

To extract the features from the raw PCAP files, first we convert the files into the Argus compatible format. Argus is a data network transaction auditing tool that categorizes and tracks network packets that match the libpcap filter expression into a protocol-specific network flow transaction model. Argus reports on the transactions that it discovers, as periodic network flow data, that are suitable for historic and near real-time processing for forensics, trending, and alarm/alerting. Different network flow-based features can be generated using Argus, which is critical in the development of IDSs. Argus can extract up to 127 network flow features. However, not all of the network features are viable. Additionally, features such as host and source IP address, and ports, are not included, so it can be easily deceived in real-world scenarios.

In this research, 25 features/attributes, shown in Table 2, are extracted from both malicious and normal traffic. These attributes consist of network flow information, including their statistical properties as well. The final CCD_IDSv1 dataset is in CSV format for

evaluation. The dataset is labeled in two different ways: for anomaly detection and threat or multi-attack classification. Anomaly detection is binary classification, so the dataset is labeled into two classes: normal and attack. For threat classification, each different attack is labeled, including normal usage for multi-class classification.

**Table 2.** Attribute's description of CCD_IDSv1 dataset.

| SN | Attributes | Description |
|----|-----------|-------------|
| 1 | Dur | Total record duration |
| 2 | Runtime | Total active flow run time generated through aggregation, the sum of the records' duration |
| 3 | Idle | Time since the last packet activity |
| 4 | Mean | Average duration of aggregated record |
| 5 | Stddev | Standard deviation of aggregated duration times |
| 6 | Sum | Total accumulated durations of aggregated records |
| 7 | Min | Minimum duration of aggregated records |
| 8 | Max | Maximum duration of aggregated records |
| 9 | Proto | Transaction protocol |
| 10 | Cause | Argus record cause code (start, status, stop, close, and error) |
| 11 | Pkts | Total transaction packet count |
| 12 | Spkts | Source to destination packet count |
| 13 | Dpkts | Destination to source packet count |
| 14 | Bytes | Total transaction bytes |
| 15 | Sbytes | Source to destination transaction bytes |
| 16 | Dbytes | Destination to source transaction bytes |
| 17 | Load | Bits per second |
| 18 | Sload | Source bits per second |
| 19 | Dload | Destination bits per second |
| 20 | Rate | Packets per second |
| 21 | Srate | Source packets per second |
| 22 | Drate | Destination packets per second |
| 23 | Offset | Record byte offset in file or stream |
| 24 | Smeansz | Mean of the flow packet size transmitted by the source |
| 25 | Dmeansz | Mean of the flow packet size transmitted by the destination |

### 3.1.4. Training and Testing Dataset

Furthermore, we balanced the dataset by under-sampling so that all class labels have the same number for both anomaly detection and threat classification. This is required to reduce biases towards any class with high representation. The total number of data was around 7.6 million. After balancing the dataset, it is divided into 80% training and 20% test datasets. Table 3 shows the total number of training and test data for binary classification (anomaly detection) and multi-class classification (threat classification).

**Table 3.** Training and test datasets for both anomaly detection and multi-attack classification after under-sampling.

| Dataset | Train | Test |
|---------|-------|------|
| Anomaly Detection | 712,792 | 177,532 |
| Multi-Attack Classification | 26,538 | 6420 |

### 3.2. ML Models

In this research, we performed a spot check with different ML models and selected a few for further study. ML models such as RF [18], KNN [19], XGBoost [20], and classification and regression trees (CART) [21] were the final models used for the study.

KNN [20] is a non-parametric supervised learning algorithm. Initially, the number of neighbors is assigned as K. It then predicts the class of a new point based on the nearest distance (Euclidean, Manhattan, or Minkowski distance measure function) between the

new point and the training samples. RF [18] incorporates a large number of decision trees working collectively as an ensemble. The collective votes from all individual trees determine the final decision or class prediction. XGBoost [21] is a highly efficient gradient boosting algorithm that improves upon gradient boosting machines through system optimization and improvement in algorithms. Time cost has also been improved through parallelization, distributed computing, out-of-core computing, and cache optimization. These features enable multi-machine training for the optimization of data structures to achieve the best global minimum and run time. CART [22] can be used as a classification tree that is not significantly impacted by outliers in the input variables and can incorporate the same variables more than once in different parts of the trees. The tree grows from the root node and splits at each node, while the leaf nodes provide the output variable. The decision tree stopping criteria, as pointed out by Zhang [22], is that each leaf node represents a single class by attaining homogeneity within prespecified training counts.

### 3.3. DL Models

In this research, we use CNN [23] and long short-term memory (LSTM) [24], used in our previous research [25], as base DL models. The embedding layer, which is generally used in natural language processing [26], is used for encoding. The embedding layer takes the input and transfers encoded data to DL models [27]. It is initialized with random weights, which are optimized with every epoch. Due to its dynamic nature, embedding performs better than static encodings such as one-hot encoding [27].

Firstly, for CNN, output from the embedding layer is fed into a first 2D convolutional layer with filter size $2 \times 2$. The output from the first convolutional layer is fed into another 2D convolutional layer with 128 filters (size $2 \times 2$). The dropout layer is used to minimize overfitting and maximize generalization. Next, a third 2D convolutional layer is used. Thereafter, a 2D max-pooling layer followed by another 2D convolutional layer is used. Then, after flattening, dense layers with four hidden layers are used. Adam [28] was used as an optimizer for the CNN architecture; it uses adaptive learning rates to calculate individual learning rates for each parameter. Softmax is used as an activation function. It assigns probabilities to each class that sum up to one.

The detailed parameter description for the CNN model is shown in Table 4. Learning rate was set to default 0.001. Furthermore, due to decent size of our dataset, we were able to utilize higher number of filters for extraction of more information from our input. Standard dropout rate was chosen throughout to achieve highly generalizable model. Higher tensor cores in our GPU enabled us to use higher batch size, but we limited to 512 to limit the possibility of missing global minima during gradient decent. Finally, Checkpointer function was used to extract best model based on highest validation accuracy and lowest loss.

**Table 4.** Parameter description of CNN model with embedding layer.

| Parameters | Settings |
|---|---|
| Learning Rate | 0.001 |
| Batch Size | 512 |
| Dropout | 0.4 |
| Conv2d_1 filter (filter size) | 64 ($2 \times 2$) |
| Conv2d_2 filter (filter size) | 128 ($2 \times 2$) |
| Conv2d_3 filter (filter size) | 256 ($2 \times 2$) |
| MaxPooling2d_1 | $2 \times 2$ |
| Conv2d_4 filter (filter size) | 128 ($2 \times 2$) |
| MaxPooling2d_2 | $2 \times 2$ |
| Dense_1 | 768 |
| Dense_2 | 256 |
| Dense_3 | 128 |
| Dense_4 | 64 |
| Output layer activation function | Softmax |
| Checkpointer | Best validation accuracy |

For LSTM, similar to CNN, output from the embedding layer is fed into two consecutive LSTM layers followed by dropout layers. Then, the output is fed into the dense layers, with the final output at the end. A model checkpoint function is used for both models to extract the best model out of all the epochs based on the validation dataset. The detailed parameter description for the LSTM model is shown in Table 5.

**Table 5.** Parameter description of LSTM model with embedding layer.

| Parameters | Settings |
|---|---|
| Learning rate | 0.001 |
| Batch size | 512 |
| LSTM layer 1 memory units | 128 |
| LSTM layer 2 memory units | 64 |
| LSTM layer 2 dropout | 0.5 |
| Dense layer 1 | 128 |
| Dropout | 0.4 |
| Dense layer 2 | 64 |
| Dropout | 0.4 |
| Output layer activation function | Softmax |
| Checkpointer | Best validation accuracy |

### 3.4. Ensemble Model

The ensemble model combines different models trained on the different datasets or with different algorithms to improve performance. In this research, we develop two ensemble models: Ensemble-CNN-10 and Ensemble-CNN-LSTM. Ensemble-CNN-10 combines ten different models trained during 10-fold cross-validation using Stacking ensemble [29], as shown in Figure 2. The stacked ensemble uses a meta-learning algorithm to find the best combination of these models. In this model, we used a neural network as a meta-learning algorithm. We can further optimize the model by changing the meta-learning algorithm as well.

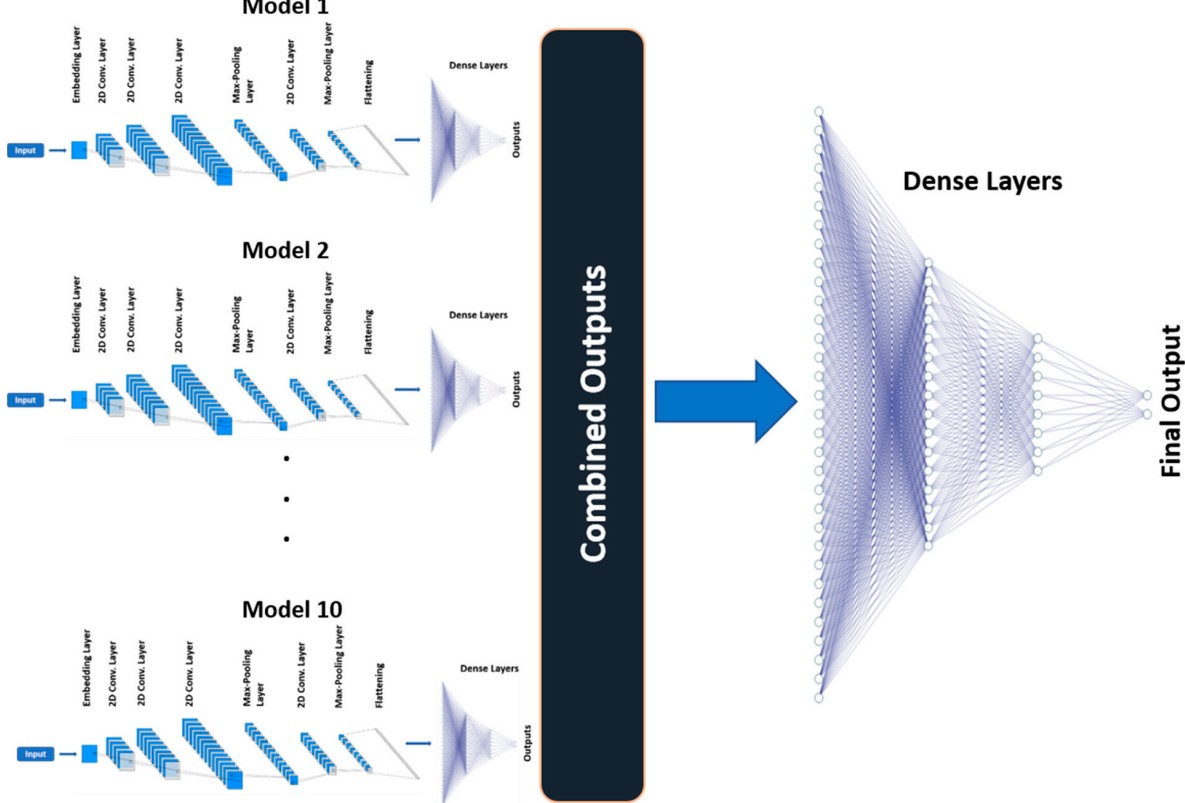

**Figure 2.** Stacking ensemble model combining 10 models from 10-fold cross-validation.

The second model, Ensemble-CNN-LSTM, as shown in Figure 3, combines CNN and LSTM models trained on the same dataset using the Stacking ensemble used before in the Ensemble-CNN-10 model. Both of the aforementioned models give high flexibility for optimization. With further increase in the number of features and complexity with the addition of more attacks, a highly optimized and versatile model is required.

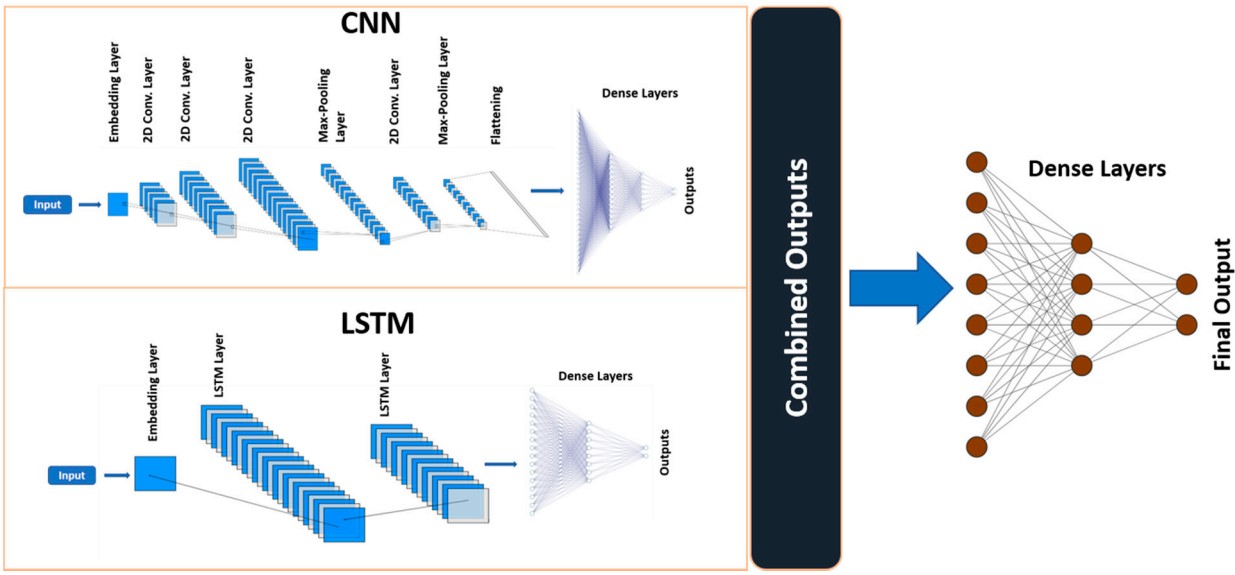

**Figure 3.** Stacking ensemble model combining CNN and LSTM models.

### 3.5. Performance Metrics

In this research work, 10-fold cross-validation was used to evaluate the performance of the model and to determine its generalizability. In 10-fold cross-validation, the data is partitioned into ten equal parts. Then, one part is left out for validation, and training is performed on the remaining nine parts. This process is repeated until all parts have been used for validation.

Confusion matrix, precision, recall, and F1-score were used as performance metrics. For binary classification, the dimension of the confusion matrix is $2 \times 2$, and for multi-class classification with six classes, the dimension is $6 \times 6$. The diagonal of the matrix gives the counts of true predicted values. It consists of true-positive ($TP$), false-positive ($FP$), true-negative ($TN$), and false-negative ($FN$). Furthermore, time cost has also been evaluated for different models to determine their efficiency. The following metrics are used:

$$\text{Accuracy} = \frac{TP + TN}{TP + TN + FP + FN} \times 100 \tag{1}$$

$$\text{Precision} = \frac{TP}{TP + FP} \times 100 \tag{2}$$

$$\text{Recall} = \frac{TP}{TP + FN} \times 100 \tag{3}$$

$$\text{F1 Score} = \frac{\text{Precision} \times \text{Recall}}{\text{Precision} + \text{Recall}} \times 2 \tag{4}$$

### 4. Results

In this section, we present the results for anomaly detection and multi-attack classification by various implemented ML and DL models. Model benchmarking and feature importance are also studied.

### 4.1. Anomaly Detection

Firstly, different ML models and DL models were applied for the analysis of the CCD-IDSv1 dataset with two classes: intrusion (includes all attacks) and normal usage. The dataset was divided into training and test sets with an 80:20 ratio. Ten-fold cross-validation was applied in the training dataset to determine the robustness and overall generalization of the model. The results are shown in Table 6. Both ML and DL models were able to achieve 100% mean accuracy with almost zero standard deviation. CART achieved 100% mean accuracy in 10-fold cross-validation with the lowest training time of 18.73 s out of all the methods. Of the two DL models, CNN had a shorter training time although still far longer than the ML models.

**Table 6.** Performance metrics for anomaly detection for 10-fold cross-validation with standard deviation and independent test.

| Models | 10-Fold Cross-Validation | | Independent Test | | | | |
|---|---|---|---|---|---|---|---|
| | Mean Accuracy (%) | Execution Time (s) | Accuracy (%) | Precision | Recall | F1-Score | Training Time (s) |
| KNN | 100 ± 0.00 | 167.92 | 99.99 | 1.00 | 1.00 | 1.00 | 4.96 |
| RF | 100 ± 0.00 | 872.46 | 100 | 1.00 | 1.00 | 1.00 | 61.85 |
| XGBoost | 100 ± 0.00 | 724.48 | 100 | 1.00 | 1.00 | 1.00 | 41.54 |
| CART | 100 ± 0.00 | 18.73 | 100 | 1.00 | 1.00 | 1.00 | 0.99 |
| LSTM | 100 ± 0.01 | 3495.41 | 99.99 | 1.00 | 1.00 | 1.00 | 320.54 |
| CNN | 100 ± 0.01 | 1200.89 | 99.99 | 1.00 | 1.00 | 1.00 | 98.09 |

An independent test was performed on the 20% of the independent dataset that was not used for the training. The results are shown in Table 6. No further complex models were added to the existing ML models (KNN, RF, XGBoost, and CART) and DL models (CNN and LSTM) since these models were able to achieve perfect performance metrics in 10-fold cross-validation. CART achieved an accuracy of 100%, with a lowest execution time of 0.99 s. KNN achieved 99.99% accuracy, whereas both RF and XGBoost achieved 100% accuracy but with longer training times than CART. Both DL models—LSTM and CNN—were able to achieve 99.99% accuracy with much longer training times than the ML models. LSTM CNN training times were 320.54 and 98.09 s, respectively.

### 4.2. Multi-Attack Classification

The next step in this analysis was the multi-class classification of different attacks as well as normal usage. In total, there were six classes (five intrusion-based and one normal usage). Similar to intrusion detection, different ML models and DL models were applied for the multi-class analysis of the CCD-IDSv1 dataset. The dataset was divided into training and test sets with a 80:20 ratio, and 10-fold cross-validation was applied. The results are shown in Table 7. KNN achieved the lowest accuracy, 77.13%, with a standard deviation of 0.65. Other ML models—RF, XGBoost, and CART—were able to achieve ~95% mean accuracy. Although XGBoost attained marginally better accuracy, CART had the shortest training time of only 1.45 s. Both base DL models—LSTM and CNN—were able to achieve ~97% accuracy, with CNN slightly better with shorter training time. For multi-class classification, due to added complexity, DL models were able to perform slightly better than ML models.

**Table 7.** Performance metrics for multi-attack classification for 10-fold cross-validation with variance and independent test.

| Models | 10-Fold Cross-Validation | | Independent Test | | | | |
|---|---|---|---|---|---|---|---|
| | Mean Accuracy (%) | Execution Time (s) | Accuracy (%) | Precision | Recall | F1-Score | Training Time (s) |
| KNN | 77.13 ± 0.65 | 6.02 | 77.79 | 0.77 | 0.78 | 0.77 | 0.28 |
| RF | 95.26 ± 0.30 | 31.88 | 95.31 | 0.95 | 0.95 | 0.95 | 1.71 |
| XGBoost | 95.78 ± 0.45 | 128.93 | 95.48 | 0.96 | 0.95 | 0.95 | 7.35 |
| CART | 95.25 ± 0.36 | 1.45 | 95.36 | 0.95 | 0.95 | 0.95 | 0.08 |
| LSTM | 96.51 ± 0.73 | 515.04 | 96.18 | 0.91 | 0.88 | 0.87 | 79.39 |
| CNN | 96.90 ± 0.31 | 117.80 | 96.51 | 0.93 | 0.92 | 0.92 | 57.97 |
| Ensemble-CNN-10 | | | 95.40 | 0.93 | 0.93 | 0.93 | 153.16 |
| Ensemble-CNN-LSTM | | | 96.62 | 0.93 | 0.93 | 0.93 | 316.39 |

An independent test was performed on the 20% of the dataset that was not used for the training. The results are shown in Table 7. A similar trend was seen on the independent test as well. For the ML models, KNN had the lowest accuracy, with other ML models achieving ~95% accuracy. The CART models took the shortest training time, 0.08 s. The base DL models—LSTM and CNN—achieved better accuracies than their ML counterparts. CNN had slightly better accuracy and training cost than LSTM. To extend the research, two ensemble models were tested based on these two DL models. The Ensemble-CNN-10 model combined 10 CNN models developed from 10-fold cross-validation using stacking ensemble. However, there was no improvement in terms of performance. The ensemble was only able to improve slightly on recall and F1-score. The Ensemble-CNN-LSTM model improved overall metrics slightly, attaining a highest accuracy of 96.62%. Precision, recall, and F1-score have class dependencies due to which they were better for ML models.

### 4.3. Model Benchmarking

We implemented the ML and base DL models used in this study using the publicly available network intrusion datasets CIDDS [12,13] and CICIDS2017 [15] for model benchmarking. The results are shown in Table 8. Both ML and DL models performed well in the independent tests on both datasets, with 99% accuracy on average, as demonstrated in [25].

**Table 8.** Performance of ML and DL models on CIDDS and CIC-IDS2017 dataset.

| Models | Accuracy (%) | |
|---|---|---|
| | CIDDS External | CIC-IDS2017 |
| RF | 98.89 | 99.95 |
| XGBoost | 98.88 | 99.95 |
| CART | 99.38 | 99.97 |
| LSTM | 99.14 | 99.95 |
| CNN | 99.11 | 99.96 |

### 4.4. Feature Importance

Classical supervised ML models are white-box models, which allow us to analyze the importance of the features used. These models help us determine the effects of each feature for both binary and multi-class classification. This information enables further feature engineering to improve the performance of the model. Each feature is given a score from 0 to 1 such that its total sum is 1. In this research, feature importance was calculated using RF. In scikit-learn, we implement the feature importance as described by Breiman et al. [21]. It is based on Gini importance or means decrease impurity, defined as the total decrease in node impurity averaged over all ensemble trees. These values are relative to a specific dataset; thus, these values cannot be compared between different datasets. Hence, we have calculated feature importance for anomaly detection and multi-threat classification separately.

Feature importance for cyber anomaly detection, which is binary classification, is shown in Figure 4. The idle feature, which is the time between the packets, got the highest

score, showcasing its impact on the classification. It achieved a scaled score of 0.32. Some other features, such as offset and Dmeansz (mean of flow packet size transmitted), achieved more than 0.1 scores (10% impact).

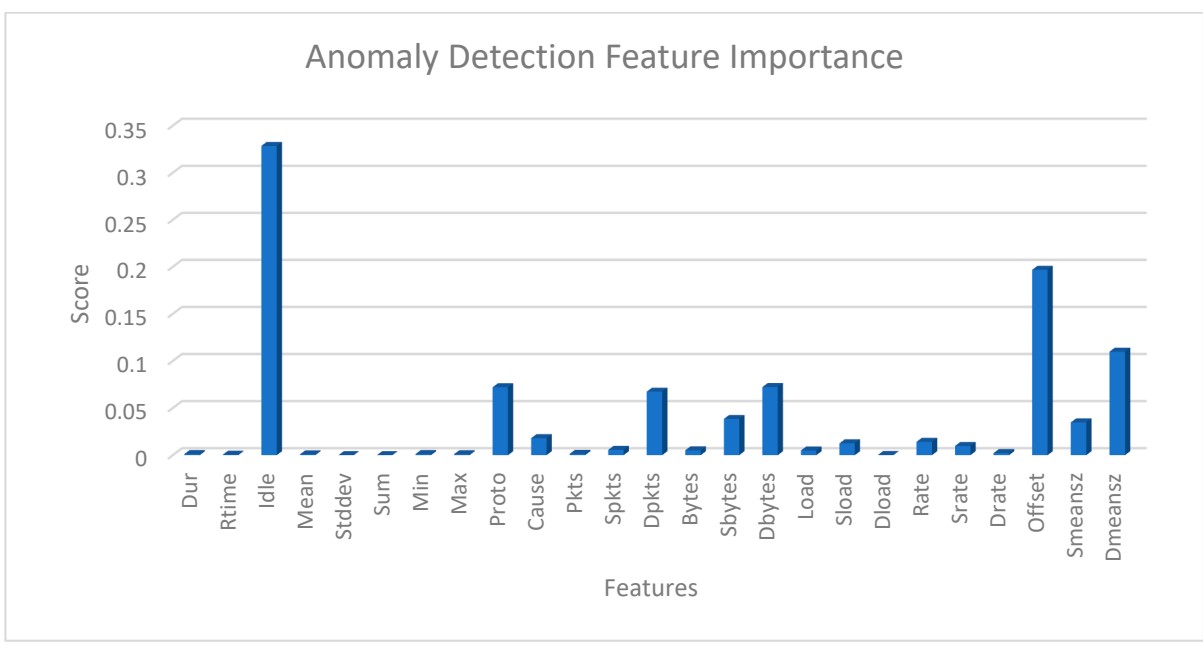

**Figure 4.** Feature importance for intrusion detection.

Feature importance for multi-threat classification, which is multi-class classification, is shown in Figure 5. The idle feature scored high in this classification as well; however, the offset feature achieved the highest score. Offset had almost 5% more impact in multi-class threat classification than in binary intrusion detection classification. Almost ten features had an impact of an around 5%. There was marginal improvement in the impact of a few features in this classification.

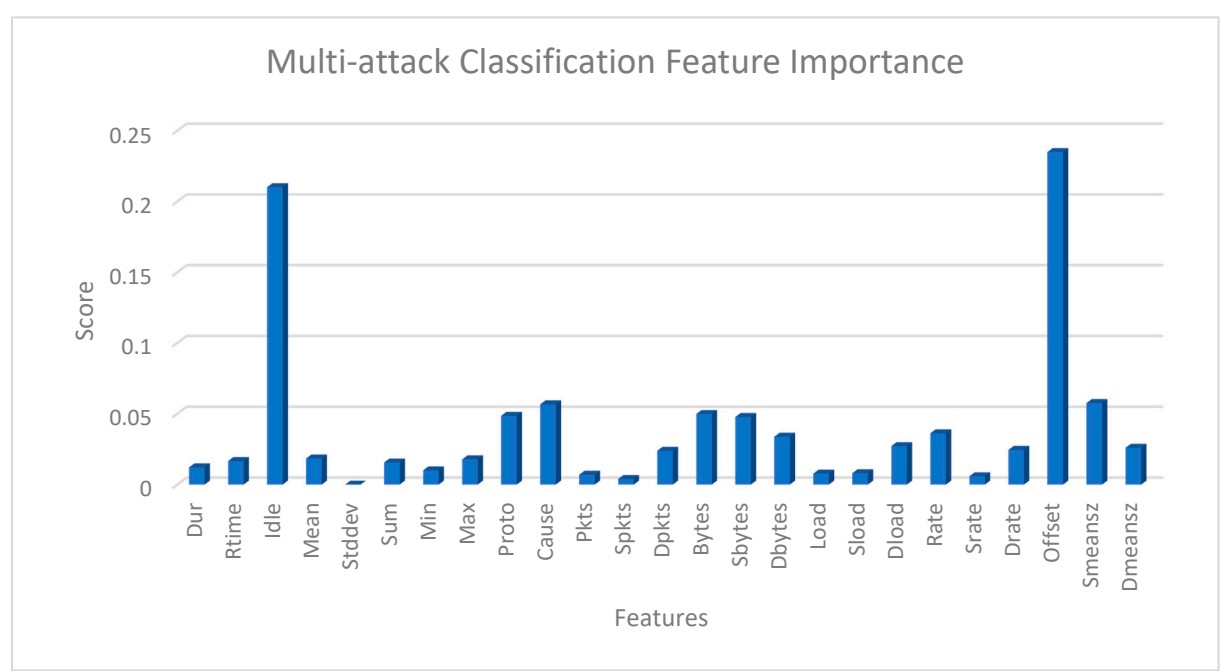

**Figure 5.** Feature importance for threat classification.

## 5. Discussion and Conclusions

In this research work, we developed a CCD-IDSv1 labeled flow-based dataset for the evaluation of an anomaly-based network IDSs. OpenStack was used to emulate a small network environment for the development of this dataset. Five different attacks, including normal usage scenarios, were implemented to collect both malicious and normal traffic, respectively. Furthermore, Argus was used to extract 25 network features. Both anomaly detection (a binary classification problem) and multi-attack classification (a multi-class classification problem) were performed. Different ML and DL models were applied on 3 datasets and used for a comparative analysis.

Overall, for the anomaly detection, ML models (KNN, RF, XGBoost, and CART) and DL models (LSTM and CNN) were able to achieve 100% accuracy in both 10-fold cross-validation as well as an independent test. The training time was shortest for CART and longest for LSTM. There were added complexities for threat classification resulting in lower accuracies for all the models compared to near-perfect performance for anomaly detection. KNN suffered most in terms of accuracy. RF, XGBoost, and CART maintained around 95% accuracy, while LSTM and CNN maintained around 96% accuracy.

Two ensemble models were developed in this research to try to improve the performance in threat classification. The first model, Ensemble-CNN-10, combined 10 CNN models developed from 10-fold cross-validation, whereas Ensemble-CNN-LSTM combined base CNN and LSTM models. Both ensemble approaches used the Stacking algorithm in combination with a neural network as a meta learner. However, Ensemble-CNN-10 was not able to improve performance. Ensemble-CNN-LSTM was able to improve performance but only slightly, attaining the highest accuracy 96.62%, in this research.

Furthermore, feature importance using RF was evaluated for both anomaly detection and threat classification. The idle attribute got the highest score for anomaly detection, whereas offset attributes achieved the highest score for threat classification. There was marginal improvement in the score for a few other features in threat classification. Training and testing of these models were carried out in a system with an Intel i7-9750 processor, 64GB RAM, and Nvidia 2080 graphics card. Training time was reduced by utilizing CUDA cores in the graphics card, which is a resource exploited by Tensorflow. In this research, higher predictability was achieved using DL models, and interpretability with lower training cost while maintaining good predictability was achieved using ML models.

For future work, the next version of CCD-IDSv1 will be developed, adding more attacks and improving upon the network environment to imitate real-life scenarios. Furthermore, different cloud servers can be considered for these environments. With improvements in datasets, further optimization of models will be required to compensate for the added complexities. Hence, further development of the ML and DL models is also required to attain better performance without sacrificing interpretability.

**Supplementary Materials:** The following are available online at https://www.mdpi.com/article/10.3390/electronics10151747/s1, Table S1: Attacks Description, Table S2: 10-fold Cross validation for different CNN configurations.

**Author Contributions:** N.T. developed dataset and ML/DL models and wrote the draft, Z.L. developed dataset and revised the draft, and A.S. developed dataset. A.E., B.G., and K.R. conceived and designed the experiment and revised the manuscript, and KR supervised the overall project. All authors have read and agreed to the published version of the manuscript.

**Funding:** This work was supported partially by CISCO Inc., a research grant. Any opinions, findings, and conclusions or recommendations expressed in this material are those of the author(s) and do not necessarily reflect the views of CISCO Inc. Equipment has been sourced through North Carolina A&T State University, Greensboro, NC 27411, US.

**Data Availability Statement:** Publicly available at https://github.com/nthapa-ds/CCD-IDSv1 (accessed on 13 May 2021).

**Conflicts of Interest:** The authors declare no conflict of interest.

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
