# Peer review of "Secure Cyber Defense: An Analysis of Network Intrusion-Based Dataset CCD-IDSv1 with Machine Learning and Deep Learning Models"

_electronics, doi:10.3390/electronics10151747_

Round 1

Reviewer 1 Report

To begin with, I liked the approach and how the experiments were conducted. Furthermore, I liked the overall distribution of the paper.

My main concern is the lack of depth of some sections. In relation to the dataset, it would be interesting to know where the traffic is collected from and how the flows are defined in terms of time length or other definition conditions.

I miss a deeper analysis of the content of the dataset from a statistical point of view, including the number of flows of each kind of traffic and other results as means of feature values for each group too.
Besides, I feel that it should include a more thorough justification of the need of this dataset. I agree with the reasons exposed about the features, but I cannot see clearly how this is addressed in the new dataset. For instance, which features not present in datasets as CIC-IDS-2017 are included in this one?
Not for this version of the dataset, but as a reference for the future versions, it would be a good idea to include attacks from the external network too.

About the ML and DL models, it would be interesting to include an explanation of the choices made on the configuration of the networks. It is more or less explained what each part does, but not why these decisions had been taken.

Also, it might be a good idea to modify the description of tables 5 and 6 because they refer to different experiments, one anomaly-based and the multi-class one.

Finally, just a couple of sentences that could be misunderstood or that might contain a minimal error.

1) "Hence, anomaly-based IDSs are preferred over signature-based IDSs, which can act against unknown attacks." This does not make it very clear if the clause after the which refers to the first or second element.

2) "Both KDD99 and NSL-KDD comparatively old and could be outdated for the present intrusion threats." Could be an "are" missing, as in "Both KDD99 and NSL-KDD are comparatively old and could be outdated for the present intrusion threats."?

Reviewer 2 Report

  1. Please add some latest references in relation to your work.
  2. Please provide justification for using the parameters in Tables 3 and 4.
  3. Did the authors use the important features to train and test the models? Please clarify how the authors have used the important features in their models.
  4. Is there any justification for choosing five specific attacks?

Reviewer 3 Report

This paper is based on the extended work of the authors on applying machine learning to intrusion detection. The major contribution is a new dataset CCD-IDSv1 which, although collected in a relative simple simulation environment, contains five attack types and more network features compared with the exiting benchmark datasets. This dataset and feature importance were evaluated using several machine learning models. Some of the proposed models were also tested on other benchmark datasets. Overall, the paper is in good quality but the novelty is a bit low. Some suggestions: (1) deep learning is a sub-field of machine learning. The authors separated DL from ML and stated that "ML models are white-box models, which allow for the interpretability of features, whereas DL models are black-box models" (lines 49-50). I understand that the authors want to highlight their DL models. But the words, especially "ML models are white-box", are not accurate. "classical supervised learning models" can be a better choice for "ML" in this paper. (2) for readers' convenience, more measurement details on feature importance can be added in section 4.4. (3) proofreading is needed. Some sentences are awkward. e.g., lines 111-112, "Furthermore, we develop an ensemble model to improve IDSs performance. Furthermore, feature importance is also studied for the CCD-IDSv1 dataset."

Round 2

Reviewer 2 Report

The changes look fine.